# Aquatic Vegetation Loss and Its Implication on Climate Regulation in a Protected Freshwater Wetland of Po River Delta Park (Italy)

**Mattias Gaglio** [1], **Mariano Bresciani** [2], **Nicola Ghirardi** [2], **Alexandra Nicoleta Muresan** [3], **Mattia Lanzoni** [1], **Fabio Vincenzi** [1], **Giuseppe Castaldelli** [1,*] **and Elisa Anna Fano** [3]

[1]  Department of Environmental and Prevention Sciences, University of Ferrara, Via Borsari 46, 44121 Ferrara, Italy; gglmts@unife.it (M.G.); mattia.lanzoni@unife.it (M.L.); fabio.vincenzi@unife.it (F.V.)
[2]  CNR-IREA, Optical Remote Sensing Group, Via Bassini 15, 20133 Milano, Italy; bresciani.m@irea.cnr.it (M.B.); ghirardi.n@irea.cnr.it (N.G.)
[3]  Department of Life Sciences and Biotechnology, University of Ferrara, Via Borsari 42, 44121 Ferrara, Italy; alexandranicoleta.muresan@unife.it (A.N.M.); fne@unife.it (E.A.F.)
*   Correspondence: ctg@unife.it

**Abstract:** Aquatic vegetation loss caused substantial decrease of ecosystem processes and services during the last decades, particularly for the capacity of these ecosystems to sequester and store carbon from the atmosphere. This study investigated the extent of aquatic emergent vegetation loss for the period 1985–2018 and the consequent effects on carbon sequestration and storage capacity of Valle Santa wetland, a protected freshwater wetland dominated by *Phragmites australis* located in the Po river delta Park (Northern Italy), as a function of primary productivity and biomass decomposition, assessed by means of satellite images and experimental measures. The results showed an extended loss of aquatic vegetated habitats during the considered period, with 1989 being the year with higher productivity. The mean breakdown rates of *P. australis* were 0.00532 d$^{-1}$ and 0.00228 d$^{-1}$ for leaf and stem carbon content, respectively, leading to a predicted annual decomposition of 64.6% of the total biomass carbon. For 2018 the carbon sequestration capacity was estimated equal to 0.249 kg C m$^{-2}$ yr$^{-1}$, while the carbon storage of the whole wetland was $1.75 \times 10^3$ t C (0.70 kg C m$^{-2}$). Nonetheless, despite the protection efforts over time, the vegetation loss occurred during the last decades significantly decreased carbon sequestration and storage by 51.6%, when comparing 2018 and 1989. No statistically significant effects were found for water descriptors. This study demonstrated that *P. australis*-dominated wetlands support important ecosystem processes and should be regarded as an important carbon sink under an ecosystem services perspective, with the aim to maximize their capacity to mitigate climate change.

**Keywords:** *Phragmites australis*; carbon storage; carbon sequestration; remote sensing; vegetation indexes; Po river delta; wetland management; climate change

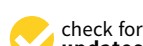



## 1. Introduction

Wetlands are important environmental components for human well-being and sustainable development since provide a multiplicity of ecosystem services and support aquatic biodiversity [1,2]. Nonetheless, they are affected by several pressures and impacts deriving from human activities and climate change, which cause their loss and the dramatic degradation of their environmental quality worldwide [3,4]. According to the Millennium Ecosystem Assessment [5], about the 50% of inner wetlands were lost during the XX century in Europe, Northern America and Australia, while habitats and species they host are among those that suffered the most negative impacts. In order to halt this trend, different policy measures were adopted at international and local levels. The Ramsar Convention (1971) was the first international initiative specifically focused to wetlands protection and had an

important pulse since the early 1980s with the adoption of the 'wise use' principle, according to which ecosystem properties should be maintained and restored [6]. EU legislation provided further opportunities to strengthen wetland conservation with the establishment of the Natura2000 network, under the framework of EU Birds and Habitat Directives. Additionally, wetlands have undergone local protection through the widespread establishment of National and Regional parks, as well as other local protection initiatives. Despite the number of protection initiatives undertaken during the last decades, the decline of environmental quality of wetlands is still ongoing [7,8]. Particularly, aquatic vegetation loss represents one the most significant forms of ecological deterioration in fresh and brackish wetlands, which can be observed in terms of both reduction of vegetated area and net primary production (i.e., decreased biomass production per spatial unit) [9–11]. Aquatic vegetated habitats, including emergent, floating and submerged macrophytes, are key functional features, being essential energetic inputs for the whole ecosystem. The degradation of their organic matter is an important part of the aquatic carbon cycle, supporting food and debris chains and ensuring the storage of carbon that was previously sequestered from the atmosphere during the vegetative period. Many studies described leaf breakdown dynamics under different conditions and influencing factors, including variations of salinity, surrounding land use, water depth and species [12–16], but the consequences of aquatic macrophytes loss in terms of reduced organic inputs to support wetland processes and functions are often overlooked. Additionally, aquatic vegetation plays an important role in supporting biodiversity and water quality. In fact, it provides habitat for fishes and invertebrates, as well as nesting sites for several aquatic birds, often of high conservation or economic interest [17,18]. Aquatic plants release oxygen along water column and remove nutrients by promoting the presence of biofilms on their water-root surfaces [19,20].

Due to their importance for wetland functionality and conservation, aquatic macrophytes are considered by the EU Water Framework Directive (WFD, 2000/60/EC) as indicators of the ecological status of water bodies. Therefore, the monitoring of aquatic vegetation is fundamental to inform environmental managers on the ecological conditions of wetlands and link them to the ecosystem functionality. Remote sensing technics rapidly enhanced their potential to assist wetland monitoring during last years to estimate aquatic vegetation extension, biomass, biophysical and biochemical parameters, biodiversity and exotic species presence [21,22]. At present, a variety of optical and radar remotely sensed images are available for mapping wetland vegetation at different levels by a range of Unmanned Aerial Vehicle (UAV), airborne and space-borne sensors from multi-spectral to hyperspectral sensors, with different temporal and spatial resolution from daily to weeks or months and from meter to hundreds of meters, respectively (e.g., [23,24]). Multi-temporal and spatial remote sensing images have also been positively applied to characterize aquatic vegetation cover and temporal processes [25–27].

The common image analysis techniques used in mapping wetland vegetation include digital image classification (e.g., [28]), various change analysis methods and spectral vegetation index (e.g., [29]).The archived moderate resolution Landsat time-series data provide an exclusive opportunity to detect and identify wetland changes as a result of the extensive historic imagery library—free of cost, in fact. Landsat, since 1972, is the longest running uninterrupted Earth observation program [30] and the Landsat archive was the first to offer global imagery at 30 m resolution without restriction in a free and open manner [31]. Landsat images are used in different work to maps the wetlands and their changes over time (e.g., [32,33]). The newly-launched (23 June 2015) and free-available Sentinel-2 (S2) sensor offers a new opportunity to integrate and increase the Landsat dataset, with the advantage of having a higher spatial resolution (10 m of S2 compared to 30 m of Landsat). For example, Pinardi et al. [34] and Bhatnagar et al. [35] mapped spatial and temporal dynamics of vegetation communities inside wetlands using S2 imagery.

While different studies described aquatic vegetation losses worldwide and many others measured breakdown rates in water bodies, there is an existing gap in linking the consequences of ecological degradation of wetlands on carbon cycle over time. Given

the role of aquatic vegetation biomass as energetic input for aquatic biota and wetlands functioning, this connection can assist in estimating the degradation of ecological functioning of wetlands during last decades and in highlighting the role of vegetated habitats in environmental conservation. This work aims to: (i) describe the loss of emergent vegetation in a protected freshwater wetlands of the Po river delta Park (Northern Italy) during a long-term period (1985–2018), chosen as case study of a general pattern occurred in the whole delta in the same period, accounted as net aboveground production measured by means of field calibrated satellite images and (ii) estimate the consequences on carbon cycle by considering both estimated aboveground production, observed breakdown rates and abiotic conditions.

## 2. Materials and Methods

### 2.1. Study Area

Valle Santa is a freshwater wetland of 250 ha which is part of the Valli di Argenta, a 3 wetlands-system located in the province of Ferrara (Northern Italy) at 6–9 m a.s.l. (Figure 1). The wetland system was declared as site of national interest in 1976 according to the Ramsar Convention, recognized as Special Protected Areas (SPA) in 2006 and recently designated as Special Areas of Conservation (SAC) in 2019. Although part of river Reno basin, the wetlands are included within the River Po delta park, as a remnant environment of the ancient landscape of the river Po delta. In fact, the river Po delta area was subjected to extensive reclamations since the end of 19th century to the 1969 that drastically reduced the original marshes and inner wetlands in the region [8].

From a hydraulic point of view, the Valle Santa wetland is devoted to a water storage function. The water input deriving from the Idice stream is stored in the wetland during peak events to prevent floods in the nearby agricultural lands, while it is released when needed for irrigation. The wetland is dominated by monospecific *Phragmites australis* habitats with a sparse presence of floating plants (*Nuphar lutea*) and sparse trees on inner dry zones [36]. This vegetation provides nesting habitat, food and shelter to a large number of birds, amphibians and invertebrates with high conservation value and is a key source of organic matter for the whole ecosystem. Unfortunately, aquatic vegetation suffered important losses during the last decades due to different contributory factors, such as the variation of hydraulic regimes and the grazing activity related to the increasing abundance of common carps. As a consequence, the wetland faced a reduction of ecological functionality which could seriously affects ecosystem services and biological conservation.

### 2.2. Loss of Aquatic Vegetation and Aboveground Biomass

Due to their dominance on the total biomass of the wetland, *P. australis* was selected as model plant species for the analysis. In fact, the contribution of other macrophytes (e.g., *N. lutea*) and trees in terms of total biomass is assumed to be negligible. Aboveground biomass loss was estimated by the processing and comparing satellite images at different dates after calibration and validation routines based on field measures. The dates considered were: 1985, 1989, 1997, 2010, 2016, 2017 and 2018. The 1985 represents the first date covered by satellite images with suitable quality. The most recent dates (2016, 2017 and 2018) were selected for capturing a more reliable description of current situation. Other dates were selected for covering coherently the period of analysis, according the availability of cloud-free images. Aboveground Ash Free Dry Biomass (AFDB) per spatial unit (g AFDB m$^2$) was measured in 7 georeferenced sampling sites in September 2017 for the calibration of biomass predictions (Figure S1). *P. australis* plants were cut on a 1 m$^2$ surface and brought to the laboratory for the measure of AFDB, and carbon content. Biomass samples were dried at 50 °C for 72 h and subsequently put in a muffle furnace at 375 °C for 3 h to obtain dry weight and ash content, respectively. Carbon content (%) was assessed analyzing 15 g samples of biomass with TOC-V SHIMADZU (solid module SMM-5000A), previously shredded with a 0.2 mm mesh mill (FRITSCH, pulverisette 14).

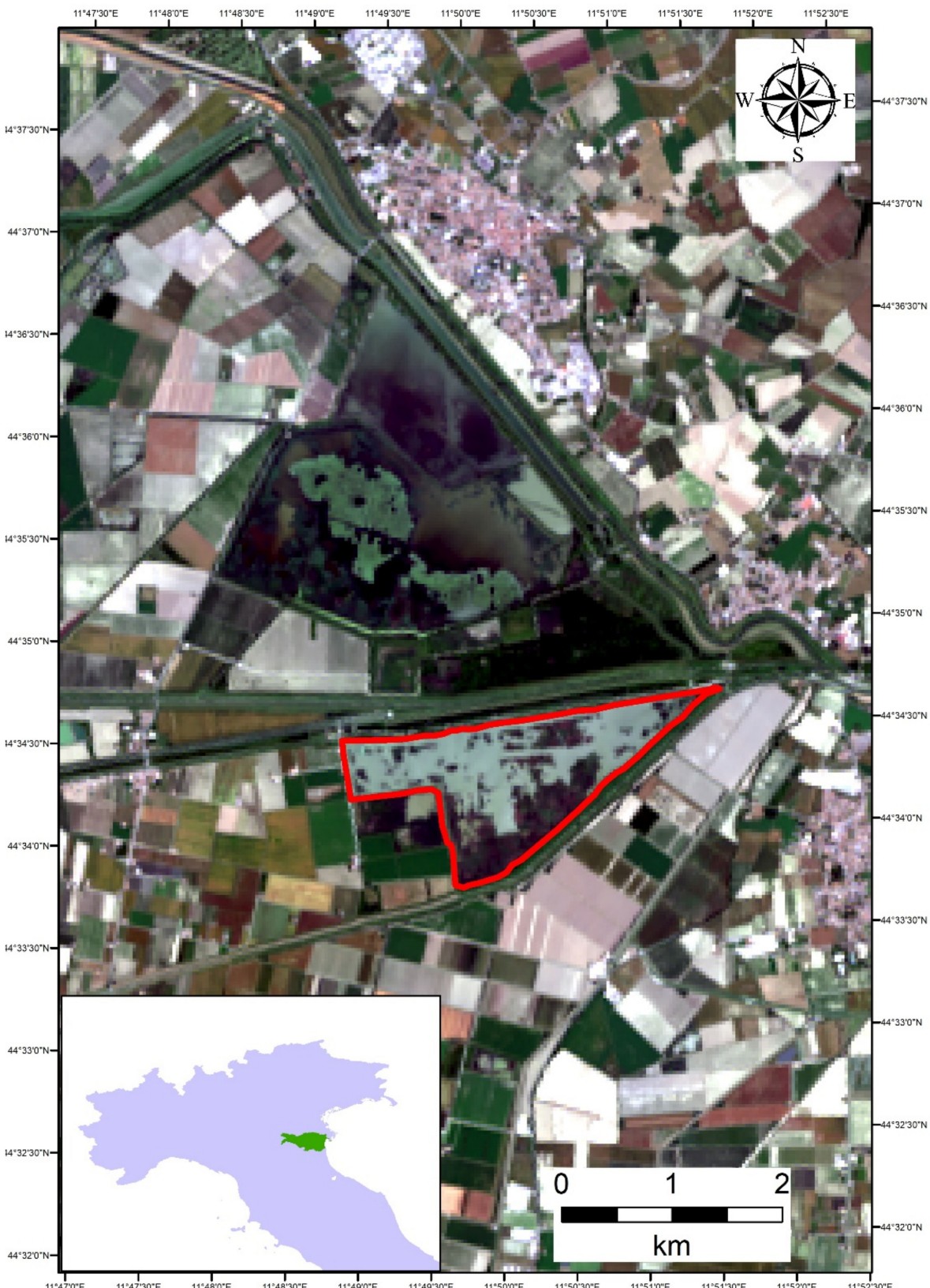

**Figure 1.** Valle Santa wetland (highlighted in red) and Ferrara province in Italy (in the left back panel).

Abiotic conditions were measured by means of water descriptors at each sampling date to test possible effects due to vegetation presence: $NH_4^+$, $NO_2^-$, $NO_3^-$, $PO_4^{3-}$, suspended sediments (total, inorganic and organic fractions), $O_2$, pH and temperature. Water

temperature and dissolved oxygen concentration were measured using a multiparameter probe (YSI Model 85), pH was measured with pH meter (Hanna Instruments HI 9026). $NH_4^+$, $NO_2^-$ and $PO_4^{3-}$ were measured using the Bower and Holm-Hansen protocol [37]. $NO_3^-$ was measured with automatic colorimeter method using AutoAnalyser II [38,39]. Total suspended solids (TSS) were quantified filtering water (GFF Whatman filters 0.7 µm porosity) using a vacuum filtration system. The inorganic (ISS) and organic (OSS) fractions of TSS were also measured by drying samples at 105 °C for 72 h and putting them in a furnace at 375 °C for 3 h.

Based on research literature of different vegetation index applied to multispectral optical satellite data, we selected six vegetation indexes used for mapping wetland vegetation and calculated from seven satellite scenes previously corrected to remove atmospheric disturbances. The images were selected from Landsat database (https://earthexplorer.usgs.gov/ accessed on 31 August 2021) and Sentinel-2 Open Access Hub (https://scihub.copernicus.eu accessed on 31 August 2021) for the months of September-November according the availability of cloud free scenes (Table 1). The satellite images were radiometrically calibrated and converted to surface reflectance after atmospheric correction performed with the 6SV code (Second Simulation of the Satellite Signal in the Solar Spectrum—Vector, [40]). We selected the Continental aerosol model available for 6SV code and the values of Aerosol Optical Thickness (AOT) were retrieved (where available) from daily MODIS products and Ozone concentration from OMI-Aura (Ozone Monitoring Instrument), via NASA Giovanni interface [41]. The different vegetation index was mentioned in the Table 2. Their performances on predicting aboveground biomass were tested using the fittest regression model. The Chlorophyll Index Green (CIGreen) was found to be the best predictor ($R^2$ = 0.827, $p < 0.01$) and was therefore chosen for calibration and validation routine with field measures (Table 2). Notably, all the vegetation indices significantly fitted field data and could be suitable for calibration.

**Table 1.** Dates and satellite source of the processed satellite images. The Landsat-5 images have a spatial resolution of 30 m, the Sentinel-2 images have a spatial resolution of 10 m.

| Date | Satellite |
|---|---|
| 9 October 1985 | Landsat 5-TM |
| 4 October 1989 | Landsat 5-TM |
| 24 September 1997 | Landsat 5-TM |
| 12 September 2010 | Landsat 5-TM |
| 15 November 2016 | Sentinel 2-MSI |
| 21 September 2017 | Sentinel 2-MSI |
| 31 October 2018 | Sentinel 2-MSI |

**Table 2.** Spectral vegetation indexes tested and fittest models results. CIGreen showed the highest $R^2$ value.

| Vegetation Index | Reference | Fittest Model | | | |
|---|---|---|---|---|---|
| | | $R^2$ | $p$-Value | Model Type | Equation |
| Chlorophyll Index Green (CIGreen) | [42] | 0.837 | <0.01 | Exponential | $y = e^{(6.29595+0.660924x)}$ |
| MERIS Terrestrial Chlorophyll Index (MTCI) | [43] | 0.803 | <0.01 | Squared | $y = 595.21 + 4747.09x^2$ |
| Modified Chlorophyll Absorption in Reflectance Index (MCARI) | [44] | 0.792 | <0.01 | Exponential | $y = e^{(6.32046+9.87445x)}$ |
| Normalized Difference Vegetation Index (NDVI) | [45] | 0.731 | 0.014 | Exponential | $y = e^{(5.82568+2.94848x)}$ |
| Normalized Difference Aquatic Vegetation Index (NDAVI) | [46] | 0.709 | <0.01 | Exponential | $y = e^{(5.8476+2.85724x)}$ |
| Enhanced Vegetation Index (EVI) | [47] | 0.706 | 0.018 | Exponential | $y = e^{(5.89928+5.66156x)}$ |

### 2.3. Breakdown Rate and Climate Regulation

The ecological functions of carbon sequestration and storage that underpin the climate regulation capacity of the wetland were assessed as a function of primary productivity and breakdown rate. The breakdown rates of stems and leaves of *P. australis* were assessed using the litterbag method [48]. A total of 72 litterbags were filled with 5 g of stems and leaves. Three replicates for both leaves and stems were collected after 7, 30, 60, 119, 182 days and brought to the laboratory for the measure of remaining ash-free dry biomass (AFDB) and carbon content, following the abovementioned procedure. Experiments were replicated in two stations (covered by vegetated habitats and bare sediment) (Figure S1) and with different mesh sizes (10 × 10 mm and 1 × 1 mm) in order to capture local variability.

The decay rates of leaves and stems were calculated according Olson (1963) [49]:

$$M_t = M_0\, e^{-kt} \tag{1}$$

where $M_t$ is the AFDB at time t, $M_0$ is the initial AFDB at the day 0, $k$ is the decay rate (days$^{-1}$) and $t$ are the number of days spent by the litterbags in water.

The capacity of the ecosystem to sequester and store carbon was estimated according the following formulas adapted from Duke et al. (2015) [50] and Gaglio et al. (2019) [18]:

$$C_{seq} = Cf \times \left( \mathrm{AFDB} - (\Delta_{stem} \times \mathrm{AFDB} \times 0.75) + \left( \Delta_{leaf} \times \mathrm{AFDB} \times 0.25 \right) \right) \tag{2}$$

$$C_{stored} = Cf \times \mathrm{AFDB} \tag{3}$$

where $C_{seq}$ is the amount of carbon that remain in the system after 1 year (C yr$^{-1}$), $C_{stored}$ is the amount of carbon stored by the system, AFDB is the aboveground biomass expressed as ash free dry biomass, $Cf$ is the carbon fraction of aboveground biomass, $\Delta_{stem}$ and $\Delta_{leaf}$ are the annual estimated fraction of carbon loss of stems and leaves, respectively. A 3:1 stem/leaf ratio on the total biomass was considered. $C_{seq}$ and $C_{stored}$ were estimated for each considered date on the basis of AFDB values derived from satellite images. Their monetary value was also assessed using the global social cost of carbon. As calculated by Rickle et al. (2018) [51], 1 t of $CO_2$ generates a global social cost of 418 US$ (according to a SSP2/RCP60 with discount growth adjusted scenario), corresponding to 96.9 € t$^{-1}$ C (US$-€ exchange rate of 0.86).

## 3. Results

### 3.1. Aboveground Biomass over Time

Satellite images were calibrated using aboveground AFDB values, measured in 7 sample sites in November 2017, ranging from 667.7 to 2429.7 g m$^{-2}$ (Table 3). It is worth mentioning that the values reported in Table 3 are not related to net primary production (i.e., biomass production per time), rather they represent the epigean biomass currently present in the sampled site. The 1989 was the year with the highest mean AFDB value (2.49 kg m$^{-2}$), while 2016 was the lowest (0.97 kg m$^{-2}$) (Figure 2). The progressive vegetation loss over time is spatially showed in Figure 3. The peak of >15 kg AFDB m$^2$ is due to the sparse presence of terrestrial vegetation (i.e., trees) in the dryer zones.

**Table 3.** Aboveground biomass, expressed as fresh matter and Ash Free Dry Biomass (AFDB), in the 7 sampling sites used for calibration of satellite images.

| Sampling Site | Aboveground Biomass | | Coordinates | |
|---|---|---|---|---|
| | Fresh Matter (g m$^{-2}$) | AFDB (g m$^{-2}$) | N | E |
| 1 | 3219.6 | 2289.1 | 44.56938 | 11.83117 |
| 2 | 2649.6 | 1739.5 | 44.57368 | 11.81999 |
| 3 | 1639.8 | 1056.1 | 44.57347 | 11.82222 |
| 4 | 1178.1 | 715.1 | 44.57371 | 11.82299 |
| 5 | 2166.4 | 1337.3 | 44.57289 | 11.81776 |
| 6 | 1112.4 | 667.7 | 44.57372 | 11.82539 |
| 7 | 3744.5 | 2429.7 | 44.57388 | 11.82720 |

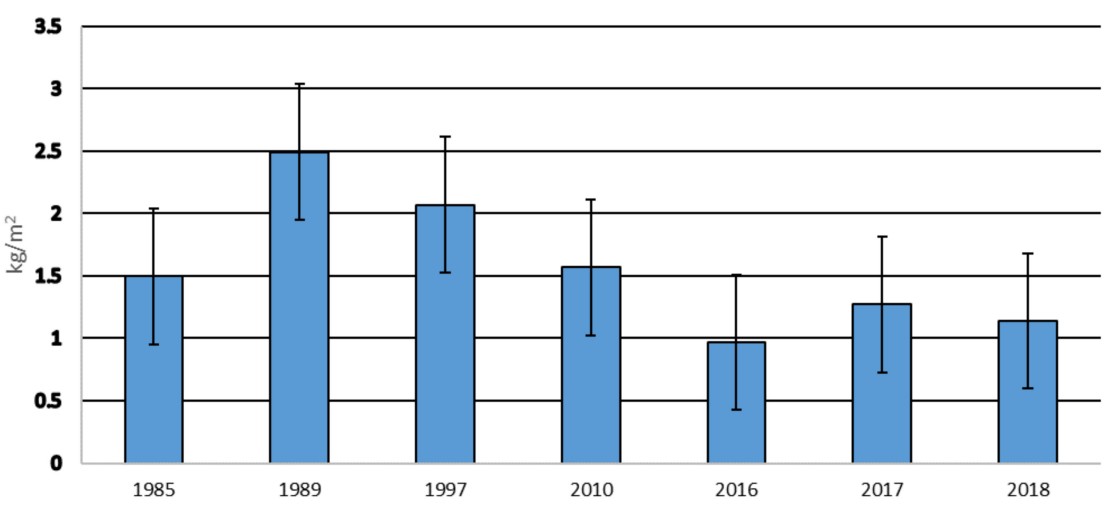

**Figure 2.** Estimates of aboveground ash free dry biomass (AFDB) per m$^2$ ($\pm$st.err.) over time (1985–2018) for each considered year.

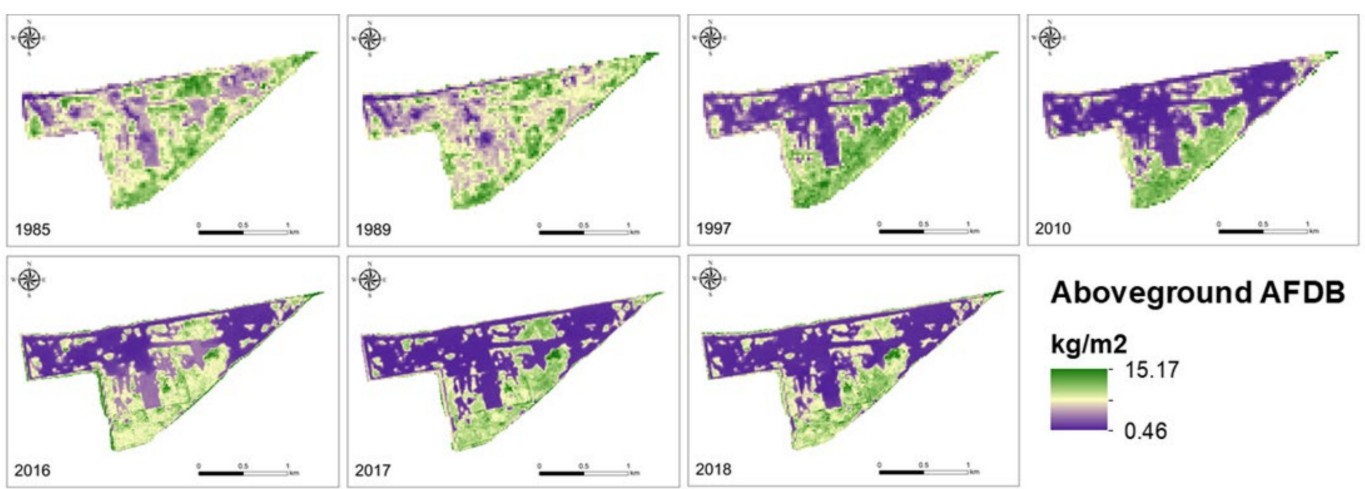

**Figure 3.** Maps of aboveground ash free dry biomass (AFDB) for each considered year.

*3.2. Biomass Decomposition*

*P. australis* bags of two different mesh sizes were collected in two sites. Since no statistical significant effect was observed for mesh sizes nor for sampling sites (Wilcoxon signed-rank test, $p > 0.05$), the data were pooled together to investigate biomass and carbon

decomposition dynamics over time. Leaves and stem decomposition was ruled by different decay rates. The observed leaf decay rates were 0.00573 $d^{-1}$ and 0.00532 $d^{-1}$ for AFDB and carbon content, respectively, while stem decay rates were 0.00268 $d^{-1}$ and 0.00228 $d^{-1}$. After 365 days the estimated remaining fraction of AFDB and carbon was 11.21% and 13.41% of initial leaves biomass and 42.64% and 44.32% of initial stem biomass, respectively.

*3.3. Carbon Storage and Sequestration*

The mean carbon fraction of *P. australis* plants resulted equal to 42.8% ($\pm$0.51 st.err) of dry biomass and 61.9% ($\pm$0.01) of AFDB. The latter value was used to calculate carbon sequestration and carbon storage of the wetland in biophysical and monetary terms (Equations (2) and (3)) (Table 4). All the values showed a clear pattern of increase between 1985–1989 and a decrease in the remaining period, with the lowest values for 2016. This trend caused a loss of 54.3% of carbon sequestration and storage. Specifically, 2085 t C stored and 736.7 of C sequestered per year were lost during the period 1989–2018. In monetary terms, the loss of ecosystem services corresponds to 202,140 € for carbon storage and 71,414 € $yr^{-1}$ for carbon sequestration.

**Table 4.** Carbon sequestration and storage provided by Valle Santa wetland and related monetary values over time.

| | | C Sequestration | | | | C Storage | | |
|---|---|---|---|---|---|---|---|---|
| Year | g AFDB $m^{-2}$ | g C $m^{-2}$ | g C $m^{-2}$ $yr^{-1}$ | t C $yr^{-1}$ | € $10^3$ $yr^{-1}$ | t AFDB | t C | € $10^3$ |
| 1985 | 1495.9 | 925.7 | 327.0 | 814.3 | 78.9 | 3724.9 | 2305.0 | 223.4 |
| 1989 | 2492.5 | 1542.3 | 544.9 | 1356.8 | 131.5 | 6206.3 | 3840.4 | 372.3 |
| 1997 | 2069.0 | 1280.3 | 452.3 | 1126.3 | 109.2 | 5151.7 | 3187.9 | 309.0 |
| 2010 | 1570.5 | 971.8 | 343.3 | 854.9 | 82.9 | 3910.5 | 2419.8 | 234.6 |
| 2016 | 967.7 | 598.8 | 211.5 | 526.8 | 51.0 | 2409.5 | 1491.0 | 144.5 |
| 2017 | 1270.4 | 786.1 | 277.7 | 691.5 | 67.0 | 3163.3 | 1957.4 | 189.7 |
| 2018 | 1139.1 | 704.9 | 249.0 | 620.1 | 60.1 | 2836.4 | 1755.2 | 170.1 |

Because of their high variability, no statistically significant differences (Kruskal–Wallis test $p > 0.05$) were observed among the three different sites, suggesting that vegetation presence has no significant effects on water conditions (Table 5). However, total suspended solids (TSS) were higher in vegetated sites, although not statistically significant ($p = 0.11$). $NO_3^-$ was the nutrient with higher concentrations in all the sampling sites.

**Table 5.** Mean values ($\pm$st.err.) of water parameters at the three sampling sites (E = water entrance; N-V = non-vegetated site; V = vegetated site).

| | Unit | E | N-V | V |
|---|---|---|---|---|
| $NH_4^+$ | mg/L | 0.14 ($\pm$0.03) | 0.07 ($\pm$0.01) | 0.13 ($\pm$0.01) |
| $NO_2^-$ | mg/L | 0.12 ($\pm$0.03) | 0.09 ($\pm$0.03) | 0.09 ($\pm$0.03) |
| $NO_3^-$ | mg/L | 4.27 ($\pm$1.11) | 2.45 ($\pm$1.00) | 2.41 ($\pm$0.93) |
| $PO_4^{3-}$ | mg/L | 0.01 ($\pm$0.001) | 0.01 ($\pm$0.001) | 0.01 ($\pm$0.001) |
| TSS | mg/L | 69.15 ($\pm$8.75) | 71.44 ($\pm$9.66) | 147.27 ($\pm$17.06) |
| ISS | mg/L | 54.18 ($\pm$7.49) | 52.70 ($\pm$7.03) | 117.18 ($\pm$14.35) |
| OSS | mg/L | 14.97 ($\pm$1.52) | 18.75 ($\pm$2.90) | 30.08 ($\pm$2.93) |
| $O_2$ | mg/L | 8.63 ($\pm$0.70) | 9.70 ($\pm$0.75) | 8.54 ($\pm$0.47) |
| pH | - | 7.30 ($\pm$0.09) | 7.60 ($\pm$0.05) | 7.60 ($\pm$0.04) |
| Temp | °C | 11.30 ($\pm$1.53) | 11.60 ($\pm$1.54) | 11.70 ($\pm$1.38) |

TSS = Total Suspended Solids; ISS = Inorganic Suspended Solids; OSS = Organic Suspended Solids.

## 4. Discussion

Die-back events of *P. australis* habitats have been well-documented in Europe but the understanding and quantification of their ecological consequences are still challenging [52–54]. The analysis highlighted the sharp decrease of plant biomass in the Valle

Santa wetland over time, mainly due to an extended reduction of *P. australis* habitats. The results revealed that this phenomenon had important implications for carbon cycle. The decrease of aboveground biomass caused the important loss of ecosystem services during the considered period, such as the climate regulation performed by carbon sequestration and storage functions. However, it has to be noted that the estimations of aboveground biomass combine values related to *P. australis* dominated habitats, no vegetated areas (i.e., open water surfaces) and, to a lesser extent, terrestrial habitats with sparse vegetation. Since the majority of the studies measured the productivity of specific freshwater habitats rather than the total biomass amount of wetlands, the comparison with other literature values may be difficult. For instance, measures of biomass production and decomposition rate of common reed beds are available for different brackish environments of the Po river delta. Mean aboveground biomass was found to be approximately 800 g m$^{-2}$ in September-October [55], while values of annual aboveground production ranged from 876 g dry matter m$^{-2}$ and 1056 g AFDB m$^{-2}$ [18]. However, the higher production values observed in this study are in line with other findings [56] and can be explained by the higher productivity of *P. australis* in freshwater ecosystems.

The decay rates of *P. australis* measured in Valle Santa wetland are coherent with other values observed in similar environments. For example, Longhi et al. (2008) [57] found that about the 40% of whole aboveground part remain undecomposed after one year, which is similar to our estimation (35.4%). As expected, stems decomposed slower than leaves, due to their lower nutrient concentration, high fiber content and highly sclerenchymatous tissues [58]. According to the classification of Petersen and Cummins (1974) [48], observed k-values of leaves fall into medium range (0.005–0.010), in line with the data of Bertoli et al. (2016) [59], while stem decay can be classified as slow (<0.005). The lack of significant differences found for mesh sizes, nor for sampling sites, suggests the absence of effects due to macroinvertebrates or local conditions. The first can be explained by the fact that litterbags were abundantly covered by muddy sediment after the first 30 days, thus limiting the action of shredder organisms. The lack of differences between sampling sites can be caused by the homogeneous conditions of the wetland, as also confirmed by water quality descriptors sampled in different wetland zones (Table 5). Other decomposition rates available in literature for Po delta vary largely according abiotic conditions. Scarton et al. (2002) [55] observed 45.4% and 50.4% undecomposed biomass for *P. australis* leaves and stems, respectively, after one year. Different decomposition rates were found by Gaglio et al. (2019) [18], equal to the 4.4% and 57.5% of their initial biomass, respectively.

When compared with other ecosystems of the Po delta area, the results demonstrate that aquatic vegetation loss is expected to harm climate regulation capacity more seriously when occuring in freshwater wetlands rather than in brackish environments. This finding has important implications for wetland managers and environmental policy. Under a climate change mitigation perspective, restoring aquatic vegetation in freshwater wetlands can be an efficient solution for sequestering and storing carbon and, because of their high plant biomass productivity, should be considered primarily to other aquatic environments in the Po delta area for improving carbon sequestration. The monetary evaluation of current and past climate mitigation service aims to quantify the economic damages of aquatic vegetation loss in terms of social costs and to inform environmental governance on the potential benefits of environmental restoration.

Although the scope of this work is limited to the consequences of vegetation loss on carbon dynamics, understanding the causes of wetlands deterioration is fundamental to halt this trend and to adopt successful measures for future restoration. In the case of Valle Santa wetland, the factors leading to the observed disappearance of *P. australis* habitats were not clearly demonstrated. Nonetheless, water level fluctuations may have affected common reed beds, influencing water and nutrient availability, as well as the presence of oxygen in the root zone [60,61]. Given the use of the wetland for water regulation purpose, particularly to serve the surrounding croplands, water depth depends on precipitations and agricultural water demand. For this reason, climate change may be an important

driver for vegetation loss, decreasing precipitations and increasing the water demand for irrigation. Additionally, further pressures may derive from the grazing activity of common carps, widely abundant in Valle Santa wetland [62], that prevent the growth of aquatic vegetation by continuously resuspending sediments from the bottom. The results of this study also contribute to the carbon source-sink dilemma of wetlands [63]. Wetlands act both as sink of carbon dioxide, by means of sequestration of carbon dioxide from the atmosphere and storage, and as natural sources of greenhouse gases emissions, especially methane. Determining the net results of these processes is a key challenge to determine whether aquatic ecosystems contribute positively or negatively to climate change. In this sense, the results presented in this study provide a quantification of the sink process, fundamental to offset the other processes leading to emissions of greenhouse gases (e.g., methanogenesis). Moreover, aquatic vegetation has a double role of sink for carbon dioxide and of avoidance of greenhouse gas emissions. In facts, the presence of *P. australis* in inundated freshwater sediments significantly attenuates methane emissions, both by reducing methanogenesis and promoting methane oxidation [64]. On the other hand, the analysis of water chemistry did not highlight any nutrient retention process due to *P. australis* presence. Even though phytodepuration function of common reed beds are widely documented [65], denitrification processes are not supported in absence of constant water fluxes [66], such as the case of Valle Santa wetland. The high concentration of $NO_3^-$ is coherent with the diffuse agricultural pollution that occur in the surrounding land and within the basin of Idice stream. The higher suspended solids observed in vegetated habitats, although not statistically significant, may be due to sediment resuspension performed by common carps and adults of *Procambarus clarkii*, which are abundantly present in the wetland and other local inland waters [62,67], rather to a trapping effect of vegetation. Therefore, under an ecosystem services perspective, the wetland is used to regulate the timing of water flows but not water quality.

Although the study successfully integrates remote sensing and field measures, some limitations should be considered for the interpretation of results. Satellite images can capture only aboveground biomass. While this will capture vegetation changes over time, the carbon stored in belowground biomass and sediment are omitted. Assessment of carbon sequestration and storage are also affected by the assumption that decomposition processes occur entirely in water and no biomass decay in terrestrial environment are considered.

## 5. Conclusions

The presence and maintenance of aquatic macrophytes are fundamental for mitigating climate change. In fact, the capacity of inland wetlands to regulate climate relies on the carbon sequestration and storing processes performed by aquatic vegetation, which are necessary to offset and possibly overcome emissions of other greenhouse gases that occur in lentic ecosystems. Therefore, the disappearance of aquatic vegetation represents a serious harm for climate and, more generally, for the provision of ecosystem services. The present study provides a quantification of carbon sequestration and storage over time, both in biophysical and monetary terms, demonstrating that the loss of *P. australis* dominated habitats caused a drastic decrease of climate regulation capacity. This phenomenon may potentially switch the role of inland wetlands from a sink to a source of greenhouse gases.

Monitoring environmental conditions of aquatic macrophytes and assessing the trend of the ecosystem functions and services that depend on their presence are critical aspects for environmental management and sustainable development. The approach adopted in this study also demonstrates the potential values of integrating remote sensing techniques and experimental measures in order to quantify the extent of vegetation loss and its consequences on climate. While remote sensing applications for environmental monitoring and assessment are rapidly evolving, their integration with well-established experimental procedures for the measure of ecological functions can provide a comprehensive understanding of ecological value of ecosystems and address environmental management. This approach finds a successful application in the case of aquatic vegetation, a key component



for the functioning of wetlands and the delivery of important ecosystem services, including climate regulation.

**Supplementary Materials:** The following supporting information can be downloaded at: https://www.mdpi.com/article/10.3390/w14010117/s1, Figure S1: Location of the sampling stations.

**Author Contributions:** Conceptualization, M.G., M.B. and E.A.F.; methodology, M.G., M.B., N.G.; software, M.B. and N.G.; validation, M.G., A.N.M. and M.L.; formal analysis, M.G., M.B., N.G. and F.V.; investigation, M.G., A.N.M., F.V. and M.L.; resources, E.A.F. and G.C.; data curation, M.G. and M.B.; writing—original draft preparation, M.G. and M.B.; writing—review and editing, M.G. and M.B.; visualization, M.G., M.B. and N.G.; supervision, E.A.F. and G.C.; project administration, G.C.; funding acquisition, G.C. and E.A.F. All authors have read and agreed to the published version of the manuscript.

**Funding:** This work was partially supported by the EU project number PGI04939 "Delta Lady-Floating cultures in River Deltas", financed by the European Interreg programme, and "CHANGE WECARE Climate cHallenges on coAstal and traNsitional chanGing arEas: WEaving a Cross-Adriatic REsponse" (ID:10043385), financed by the European Interreg Italy-Croatia programme.

**Data Availability Statement:** Data sharing not applicable.

**Acknowledgments:** The authors thank the Consorzio della Bonifica Renana, Argenta Municipality and F.I.P.S.A.S. ASD Valle Santa for allowing access to the protected wetland and for their logistic support, as well as the students of E.A. Fano, who collaborated to collected data on leaf decomposition dynamics.

**Conflicts of Interest:** The authors declare no conflict of interest.

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
