# Peer review of "Aquatic Vegetation Loss and Its Implication on Climate Regulation in a Protected Freshwater Wetland of Po River Delta Park (Italy)"

_water, doi:10.3390/w14010117_

Round 1

Reviewer 1 Report

The paper deals with the study of aquatic vegetation loss in a freshwater wetland, and the consequent effects on carbon sequestration and storage capacity. The experiment is well-designed and the mix between field and lab measurements and remote sensing is done in a very appropriate way. Besides, economic and monetary consequences are also addressed. So, I think it is good enough to be accepted, with no additional comments or suggestions

Author Response

We would like to thanks the reviewer for the positive evaluation and acceptance of the work.

Reviewer 2 Report

Comments:

I read with much interest this research manuscript on “Aquatic vegetation loss and its implication on climate regulation in a protected freshwater wetland”. The authors investigated the extent of aquatic emergent vegetation loss for the period 1985-2018 and the consequent effects on carbon sequestration and storage capacity of Valle Santa wetland, as a function of primary productivity and biomass decomposition, assessed by means of satellite images and experimental measures. 

My main concern is that this study does not have enough detailed understanding of the relationship between its process parameters.

(1) Abstract:This study pointed out the investigated period was 1985-2018, in fact, the study only selected some special years, and the reasons of these years were selected were unclear.

(2) Line 152: the acronym (such as AFDB) appeared the first time should be given the full words.

(3) Some spelling or typing mistakes should be thoroughly checked. Such as Line 160: NO3- should be NO3-, Line 165: NO3- should be NO3-, Line 168-169,the 105° and the 375° should be 105 °C and 375°C.

(4) Line 216-218:  Experiments were replicated in two stations (covered by vegetated habitats and bare sediment) and with different mesh sizes (10x10mm and 1x1mm) in order to capture local variability. But where were the two stations and the different mesh sizes plots? And why were the two stations selected for this study? Meanwhile, in Line 239-240: “measured in 7 sample sites in November 2017”,  but where were the 7 sample sites and why were 7 sample sites selected for this study? The details about the experiments should be described clearly, these information was crucial for this research.

(5)  The title is “Aquatic vegetation loss and its implication on climate regulation in a protected freshwater wetland”, but why there is no introduction to climate measurement and climate data in “Materials and methods”?re is no introduction to water quality measurement and floc testing methods in “Materials and methods”.

In conclusion, the writing of this paper needs great improvement before it is publishable.

Author Response

I read with much interest this research manuscript on “Aquatic vegetation loss and its implication on climate regulation in a protected freshwater wetland”. The authors investigated the extent of aquatic emergent vegetation loss for the period 1985-2018 and the consequent effects on carbon sequestration and storage capacity of Valle Santa wetland, as a function of primary productivity and biomass decomposition, assessed by means of satellite images and experimental measures. My main concern is that this study does not have enough detailed understanding of the relationship between its process parameters.

(1) Abstract:  This study pointed out the investigated period was 1985-2018, in fact, the study only selected some special years, and the reasons of these years were selected were unclear.

R: The vegetation loss occurred mainly during the end of 1990 and continued slightly after. Therefore, we decided to analyze images form 80s (when vegetation was in good conditions) until recent years. The specific dates were selected according images quality. The 1985 represents the first year covered by images with suitable quality. The most recent dates (2016, 2017, 2018) were selected because of their improved resolution and for capturing a more reliable description of current situation. Other dates were selected according the availability of cloud-free images, while covering coherently the period of analysis.  (see new lines 151-155).

(2) Line 152: the acronym (such as AFDB) appeared the first time should be given the full words.

R: The acronym AFDB has been now explicated (line 155).

(3) Some spelling or typing mistakes should be thoroughly checked. Such as Line 160: NO3- should be NO3-, Line 165: NO3- should be NO3-, Line 168-169,the 105° and the 375° should be 105 °C and 375°C.

R: Corrected as suggested.

(4) Line 216-218:  Experiments were replicated in two stations (covered by vegetated habitats and bare sediment) and with different mesh sizes (10x10mm and 1x1mm) in order to capture local variability. But where were the two stations and the different mesh sizes plots? And why were the two stations selected for this study? Meanwhile, in Line 239-240: “measured in 7 sample sites in November 2017”,  but where were the 7 sample sites and why were 7 sample sites selected for this study? The details about the experiments should be described clearly, these information was crucial for this research.

R: 7 is the minimum number of samplings necessary for a regression model. The 7 calibration points and the 3 sampling points (entrance, non-vegetated and vegetated sampling sites) were reported in map and added as Supplementary material (Fig.A1). All the points were located in the western part of the wetland because of their easier accessibility and the larger variability of common reed intensity.

(5)  The title is “Aquatic vegetation loss and its implication on climate regulation in a protected freshwater wetland”, but why there is no introduction to climate measurement and climate data in “Materials and methods”?re is no introduction to water quality measurement and floc testing methods in “Materials and methods”.

R: The aim of the paper was to quantify the temporal variation of two ecological functions (i.e. carbon sequestration and carbon storage) that provide the ecosystem service of global climate regulation. To better clarify this relation, we added the new lines 210-212. Since the benefit of capturing and storing CO2 occurs at global scale, the local climate was not taken into account. The water quality descriptors were the only measure of local environmental context, with particular focus on the possible effects on aquatic vegetation on improving water conditions. The methods used for their measure were described in lines 163-173.

In conclusion, the writing of this paper needs great improvement before it is publishable.

R: we thank the reviewer for the suitable remarks that contributed to improve the text.

Reviewer 3 Report

On the whole, the manuscript results are interesting and practically significant. They deserve to be published after taking into account the following comments.

  1. In my opinion, the study's main flaw is the complete lack of analysis of natural and/or anthropogenic causes of aquatic vegetation loss from 1985 to 2018. For example, what caused the maximum distribution of the studied vegetation species in 1989? It is important to analyze carbon balance changes and economic losses due to this vegetation loss in wetlands, but it is equally important to know what caused these losses. These causes should also be briefly written in the Abstract.
  2. What are the main limitations and uncertainties of your research? Give this information in a separate subsection of the manuscript.

In addition:

  1. Some information in the manuscript needs to be supported by links to sources (for example, lines 125-129, 300).
  2. Figure 1. The satellite image must be accompanied by a coordinate system. Is there no better quality satellite image of this territory?
  3. It is necessary to improve the quality (readability) of Figures 2 and 3.
  4. The English language of the manuscript needs some improvement.

Author Response

On the whole, the manuscript results are interesting and practically significant. They deserve to be published after taking into account the following comments.

R: We thank the reviewer for the positive evaluation of the work.

     In my opinion, the study's main flaw is the complete lack of analysis of natural and/or anthropogenic causes of aquatic vegetation loss from 1985 to 2018. For example, what caused the maximum distribution of the studied vegetation species in 1989? It is important to analyze carbon balance changes and economic losses due to this vegetation loss in wetlands, but it is equally important to know what caused these losses. These causes should also be briefly written in the Abstract.

R: Thanks for pointing out this important issue. Understanding the causes that led to aquatic vegetation loss is fundamental to halt this trend and to adopt successful measures for future restoration. That said, the scope of the work is to evaluate the consequences of vegetation loss rather to explore the causes. However, a short paragraph on the causes was added (new lines 351-363).

     What are the main limitations and uncertainties of your research? Give this information in a separate subsection of the manuscript.

R: The main limitations are provided in lines 386-392. Given the short extension of this part, we decided to not separate these lines from the remaining part of the discussion.

In addition:

    Some information in the manuscript needs to be supported by links to sources (for example, lines 125-129, 300).

R: References were added

    Figure 1. The satellite image must be accompanied by a coordinate system. Is there no better quality satellite image of this territory?

R: Coordinates were added to figure 1 and its quality was improved.

    It is necessary to improve the quality (readability) of Figures 2 and 3.

R: The quality of images was improved as suggested.

    The English language of the manuscript needs some improvement.

R: English language was generally revised.

Round 2

Reviewer 2 Report

I think the present manuscript will be recommended to publish.

Author Response

We thank the reviewer for the positive response and for the previous comments that helped us to improve the manuscript

Reviewer 3 Report

My final recommendations: 

  1. Your research does not represent a global overview of the issue studied but only a regional study. Therefore, I recommend supplementing the title of the manuscript as follows: "... freshwater wetland: A case study from Northern Italy."
  2. Combining Figure 1 and Figure A1 (Supplementary Material) would be much more convenient for the reader. Moreover, in the figure caption, let the reader know what you have highlighted with the red line.
  3. Line 187. R2 =82.7? Did you mean 0.827 (or 0.83)? Please also check Table 2 and the rest of the manuscript.
  4. Figure 2. Remove the outer frame of the graph. Make the grid lines black (easier to read) instead of light gray.
  5. Tables 4 and 5 (and in the text). Why do you indicate values down to hundredths and thousandths? There is no need for this. It is enough to show tenths. This primarily applies to weight units.

Author Response

We thank again the reviewer for the contribution to paper improvement. Here point-to-point replies to the specific comments.

My final recommendations: 

Your research does not represent a global overview of the issue studied but only a regional study. Therefore, I recommend supplementing the title of the manuscript as follows: "... freshwater wetland: A case study from Northern Italy."

R:  We thank the reviewer for arising this issue. We changed the title adding the location of the study: "...a protected freshwater wetland of Po river delta Park (Italy)". However, we think that this case study can be informative for global freshwater wetlands in similar conditions.

Combining Figure 1 and Figure A1 (Supplementary Material) would be much more convenient for the reader. Moreover, in the figure caption, let the reader know what you have highlighted with the red line.

R: We decided to maintain sampling locations in a separate figure. Figure S1 provides a more closer zoom of the wetland, while figure 1 shows Valle Santa wetland within the context of 3 wetlands-system and surrounding land. Caption of Fig.1 was modified as suggested.

Line 187. R2 =82.7? Did you mean 0.827 (or 0.83)? Please also check Table 2 and the rest of the manuscript.

R: Text and table 2 were corrected as suggested.

Figure 2. Remove the outer frame of the graph. Make the grid lines black (easier to read) instead of light gray.

R: Modified as suggested.

Tables 4 and 5 (and in the text). Why do you indicate values down to hundredths and thousandths? There is no need for this. It is enough to show tenths. This primarily applies to weight units.

R: We modified the tables using one decimal digit for table 4 and two decimal digits for table 5.